# Difference in prioritization of patient safety interventions between experts and patient safety managers in Japan

**Ryosuke Hayashi[1], Yosuke Hatakeyama[2], Ryo Onishi[2], Kanako Seto[2], Kunichika Matsumoto[2], Tomonori Hasegawa[2]***

1 Toho University Graduate School of Medicine, Tokyo, Japan, 2 Department of Social Medicine, Toho University School of Medicine, Tokyo, Japan

* tommie@med.toho-u.ac.jp

**Data Availability Statement:** Data cannot be shared publicly, because we have been approved to conduct secondary data analyses, but not to share the data used by the ethics committee. These

## Abstract

Although a variety of patient safety interventions have been implemented, prioritizing them in a limited resource environment is important. The intervention priorities of patient safety managers may differ from those of patient safety experts. This study aimed to clarify the difference in prioritization of interventions between experts and safety managers to better identify interventions that should be promoted in Japan. We performed a secondary data analysis of two surveys: the Delphi survey for Japanese experts and a nationwide questionnaire survey for safety managers in hospitals. Regarding the 32 interventions constituting 14 organizational-level and 18 clinical-level interventions examined in the previous studies, we assessed three correlations to examine the difference in prioritization between experts and safety managers: correlations between experts and safety managers in the three perspectives (contribution, dissemination, and priority), those between priorities of experts and safety managers at the clinical and organizational level, and those among the three perspectives in experts and safety managers. Contribution (r = 0.768) and dissemination (r = 0.689) of patient safety interventions evaluated by experts and safety managers were positively correlated, but priorities were not. Interventions with priorities that differed between experts and safety managers were identified. In experts, there was no significant correlation between contribution and priority or between dissemination and priority. For safety managers, contributions (r = 0.812) and dissemination (r = 0.691) were positively correlated with priority. Our results suggest that patient safety managers evaluated future priority based on past contributions and current dissemination, whereas experts evaluated future priority based on other factors, such as expected impacts in the future, as mentioned in the previous study. In health policymaking, promotion of patient safety interventions that were given high priority by experts, but low priority by safety managers, should be considered with possible incentives.

restrictions have been enforced by The Ethics Committee of Toho University School of Medicine. The external researchers can contact the Ethics Committee of Toho University School of Medicine regarding the use of the data but the committee does not accept applications other than Japanese language (med.rinri@ext.toho-u.ac.jp, +81-3-3762-4151). If an external researcher contacts the research team directly (tommie@med.toho-u.ac.jp (personal address of corresponding author), md20015h@st.toho-u.ac.jp (first author), health@med.toho-u.ac.jp (Department of Social Medicine, Toho University School of Medicine)), the research team members will submit reviews of external provision of data to the Ethics Committee on behalf of external researchers.

**Funding:** Initials of the authors who received each award: Hasegawa T Grant numbers awarded to each author: Health and Labour Sciences Research Grants (grant number:H29-Iryo-Ippan-004) The full name of each funder:the Japanese Ministry of Health, Labour and Welfare URL of each funder website:https://www.mhlw.go.jp/index.html The funders had no role in study design, data collection and analysis, decision to publish, or preparation of the manuscript.

**Competing interests:** The authors have declared that no competing interests exist.

## Introduction

Since the late 1990s, various activities have been introduced by governments, medical/specialty societies, accreditation bodies, and healthcare organizations in many countries to improve patient safety [1–3]. Previous studies have examined whether a certain patient safety intervention contributed to improving patient safety [4–13]. Implementing patient safety interventions requires considerable investment in resources and costs [14]. Although it is important to prioritize them in a limited resource environment [15–18], there is insufficient evidence regarding the cost of patient safety interventions [19].

The Organization for Economic Co-operation and Development (OECD) published a report titled "The Economics of Patient Safety: Strengthening a Value-Based Approach to Reducing Patient Harm at National Level" in 2017 [14]. In this report, the OECD chose patient safety experts and asked them for their "best estimate" of the cost and impact of each patient safety intervention using the Delphi method. These estimates could be based on evidence, observations, experience, difficulty in organizing, prediction of mortality, and morbidity reduction. Based on the experts' ratings of the impact and cost of patient safety interventions, the OECD extracted prioritized interventions to improve patient safety from 42 interventions at the system, organizational, and clinical levels.

By conducting a Delphi survey of Japanese patient safety experts, Hatakeyama et al. extracted patient safety interventions that should be prioritized in Japan [20]. In this study, the questionnaire consisted of 42 interventions based on the OECD report, and 6 perspectives for assessing the importance of interventions in the past (contribution), current (dissemination), and future (impact, cost, urgency, and priority). They reported that the priority of patient safety interventions had a positive relationship with the future impact and a negative relationship with current dissemination. These results suggested that experts gave high priority to interventions that were expected to be effective in the future and low priority to interventions that were already disseminated. It seemed to be important for policymakers and hospital administrators to consider the status of the medical system, the medical policies that had been taken thus far, and the circumstances that were important in setting the priority of patient safety interventions. These OECD report and previous study suggested that future priorities for interventions might be influenced not only by expected future impact, but also by past contribution and current dissemination.

Practitioners and patient safety managers are likely to provide patient safety interventions based on the needs and resources on their clinical and organizational settings. Patient safety experts might assess the priority interventions from the perspective of a healthcare system and policy. Therefore, the intervention priorities of safety managers might be different from those of experts. Previous studies have reported the priorities and determinants of patient safety interventions at the clinical level [21,22]. However, few studies have shown differences in intervention priorities between experts and safety managers. By clarifying the differences in prioritization and influential factors, we could identify interventions to be promoted through the support of healthcare system and policies.

This study aimed to investigate the differences in prioritization of patient safety interventions between experts and safety managers in Japan, focusing on the priority-setting mechanism of each group.

## Materials and methods

We performed a secondary data analysis of two surveys: the Delphi survey for Japanese experts [20] and a nationwide questionnaire survey for patient safety managers in hospitals [23]. Parts of both questionnaire items used in this study were shown in S1 and S2 Tables. All participants

of both surveys were informed about the research objective and the policy of data confidentiality and anonymity. Taking part in both surveys was voluntary, not mandatory. Therefore, we considered responses to surveys as consent to participate in the survey. Ethical approval for this secondary data analysis was obtained from the Ethics Committee of Toho University School of Medicine (No. A21063).

## Delphi survey for experts

The Delphi technique is a forecasting method that involves repeatedly asking experts to summarize their opinions [24,25]. This technique has been used to solve an array of healthcare problems ranging from those of an individual hospital or department to those of a statewide agency or state [26] and has also been used in the survey of the OECD report [14].

In the Delphi survey, the criteria for experts were to be actively involved in academic activities such as academic conferences or writing papers of patient safety. the respondents were 24 experts, including two representatives of nationwide organizations related to patient safety, five hospital administrators, seven in-hospital patient safety managers, eight researchers of patient safety, and two others in the field of patient safety. The survey was conducted over three rounds by mail (round 1) and e-mail (rounds 2 and 3), from July to October 2017. During these rounds, the results of the previous round were presented to the participants. According to the OECD report [14], the questionnaire consisted of 42 interventions at three levels in total, the system level (10 interventions), organizational level (14 interventions), and clinical level (18 interventions), and three perspectives: past contribution, current dissemination, and priority for future implementation. In each round, participants were asked to rate all 42 interventions on a 5-point Likert scale from two perspectives: dissemination (1: low to 5: high) and priority (1: low to 5: high). Ratings of past contributions were asked in round 1 only (1: small to 5: large) [20].

## Questionnaire survey for safety managers

The questionnaire survey of in-hospital patient safety managers responsible for patient safety management at each hospital was conducted to reveal the management systems and activities for improving patient safety in hospitals.

The anonymous nationwide mail survey was conducted in Japan from October to November 2017. The hospitals were selected by stratified random sampling according to the number of beds: 25% of hospitals with < 100 beds, 50% of hospitals with 100–299 beds, and 100% of hospitals with ≥ 300 beds were selected. Consequently, a questionnaire was sent to 3,215 hospitals, representing 38% (3,215/8,448) of all the hospitals in Japan.

Using the same wording as the Delphi survey for experts, respondents were asked to rate 42 interventions on a 5-point scale from two perspectives: past contribution to patient safety (contribution: 1: small to 5: large) and priority for future implementation (priority: 1: low to 5: high). They were also asked whether 14 interventions at the organizational level and 18 at the clinical level (totaling 32 interventions) were implemented in their hospitals with the wording of "at your hospital". The rate of implementation was used for the current dissemination in the questionnaire survey.

## Data analysis

We assessed the mean values of 32 interventions (Table 1), consisting of 14 organizational-level interventions and 18 clinical-level interventions in past contribution, current dissemination, and priority for future implementation. The 10 system-level interventions were not included in the questionnaire for safety managers because these interventions involved the

**Table 1. Patient safety interventions.**

| Level | Intervention |
|---|---|
| Organizational | |
| O-1 | Clinical governance frameworks and systems for patient safety |
| O-2 | Clinical incident reporting and management system |
| O-3 | Integrated patient complaint and incident reporting |
| O-4 | Monitoring and feedback of patient safety indicators |
| O-5 | Patient-engagement initiatives |
| O-6 | Clinical communication protocols and training |
| O-7 | Digital technology solutions to improve safety |
| O-8 | Human resources interventions |
| O-9 | Building a positive safety culture |
| O-10 | Infection detection, reporting, and surveillance systems |
| O-11 | Hand hygiene initiatives |
| O-12 | Antimicrobial stewardship |
| O-13 | Blood and blood product management protocols |
| O-14 | Medical equipment sterilization protocols |
| Clinical | |
| C-1 | Medication management / reconciliation protocols |
| C-2 | Transcribing error systems and protocols |
| C-3 | Smart infusion pumps and drug administration systems |
| C-4 | Aseptic technique protocols and barrier precautions |
| C-5 | Urinary catheter use and insertion protocols |
| C-6 | Central venous catheter insertion protocols |
| C-7 | Ventilator-associated pneumonia minimisation protocols |
| C-8 | Procedural / surgical checklists |
| C-9 | Operating room integration and display technology |
| C-10 | Peri-operative medication protocols |
| C-11 | Venous thromboembolism (VTE) prevention protocols |
| C-12 | Clinical care standards |
| C-13 | Pressure injury (ulcer) prevention protocols |
| C-14 | Falls prevention initiatives |
| C-15 | Acute delirium & cognitive impairment management initiatives |
| C-16 | Response to clinical deterioration |
| C-17 | Patient hydration and nutrition standards |
| C-18 | Patient identification and procedure matching protocols |

entire national healthcare system and could not be implemented in each hospital. The scores of interventions from the three perspectives were standardized into z-scores for adjusting the variables in the evaluation by experts and safety managers. As data of experts for current dissemination and future priority, we used the results of round three converged through the Delphi process for analyzing the representative perspectives of patient safety experts in Japan. We included the results of round one on past contribution, as they were only asked at round one. The perspectives of safety managers were varied depending on their own circumstances, there was no need to converge them for the analysis.

Three analyses were conducted using these scores. First, we calculated Pearson's correlation coefficients of the scores evaluated by experts and safety managers from the three perspectives. Second, we assessed the correlation between the priority scores of experts and safety managers using scatter plots. Finally, we calculated the correlations among the three perspective scores

of experts and safety managers using Pearson's correlation coefficients to clarify the factors that determine priority.

All data were analyzed using IBM SPSS Statistics version 19, and a p-value of $< 0.05$ was considered statistically significant.

## Results

### Characteristics of respondents

The respondents' characteristics are listed in Table 2. In the Delphi survey of experts, all 24 experts responded in all three rounds (Table 2A). The response rate was 18.8% (603/3,215) in the questionnaire survey of the safety managers. Safety managers from acute care hospitals accounted for 78.1%, and those from large hospital (beds $> 300$) for 37.3% (Table 2).

### Correlations between perspective scores evaluated by experts and safety managers

The correlations between the perspective scores evaluated by experts and safety managers are shown in Table 3. There were positive correlations in the score of past contribution (r = 0.768, p < 0.001) and current dissemination (r = 0.689, p < 0.001) evaluated by experts and safety managers. However, there was no significant correlation in the score of future priority (r = 0.231, p = 0.203), suggesting that experts and safety managers have different views on future priorities.

**Table 2.**

| a. Baseline characteristics (Experts). | | |
|---|---|---|
| | n | % |
| Experts | 24 | |
| Domain | | |
| Representative of nationwide organization related to patient safety | 2 | 8.3 |
| Hospital administrator | 5 | 20.8 |
| Patient safety manager | 7 | 29.2 |
| Researcher of patient safety | 8 | 33.3 |
| Other | 2 | 8.3 |
| Profession | | |
| Doctor | 15 | 62.5 |
| Nurse | 4 | 16.7 |
| Pharmacist | 2 | 8.3 |
| Others | 3 | 12.5 |
| **b. Baseline characteristics (Safety managers).** | | |
| | n | % |
| Safety managers | 603 | |
| Acute care hospital | | |
| < 100 beds | 68 | 11.3 |
| 100–299 beds | 178 | 29.5 |
| ≥ 300 beds | 225 | 37.3 |
| Chronic care hospital | | |
| < 100 beds | 29 | 4.8 |
| ≥ 100 beds | 48 | 8.0 |
| Psychiatric hospital | 46 | 7.6 |
| Other hospitals | 9 | 1.5 |

Table 3. Correlations between perspective scores assessed by experts and safety managers.

| | r | p value |
|---|---|---|
| Contribution | 0.768 | p<0.001 |
| Dissemination | 0.689 | p<0.001 |
| Priority | 0.231 | 0.203 |

Abbreviation: r = Pearson's correlation coefficients.

## Differences in the priority on patient safety intervention between experts and safety managers

We present scatter plots of scores for intervention priorities at the organizational level (Fig 1A) and those at the clinical level (Fig 1B). There was no significant correlation in the scores of priorities evaluated by experts and safety managers at either the organizational level (r = 0.212, p = 0.467) or clinical level (r = 0.352, p = 0.152).

The mean values of 32 interventions, consisting of 14 organizational-level interventions and 18 clinical-level interventions in the past contribution, current dissemination, and priority for future implementation, are shown in Table 4A and 4B. There were some interventions with different evaluations between the experts and safety managers. We defined scores > 0 as 'high' and < 0 as 'low'. The interventions that were given high priority by experts, but low priority by safety managers, were "Clinical governance frameworks and systems for patient safety" (O-1), "Patient-engagement initiatives" (O-5), and "Clinical communication protocols and training" (O-6); the interventions that were given high priority by experts, but low priority by safety managers, were "Clinical incident reporting and management system" (O-2), "Building a positive safety culture" (O-9), and "Medical equipment sterilization protocols" (O-14) in the organization level. At the clinical level, the interventions that were given high priority by safety managers, but low priority by experts, were "Central venous catheter insertion protocols" (C-6), "Procedural / surgical checklists" (C-8), "Peri-operative medication protocols" (C-10), and "Clinical care standards" (C-12); the interventions that were given high priority by safety managers, but low priority by experts were "Aseptic technique protocols and barrier precautions" (C-4), "Pressure injury (ulcer) prevention protocols" (C-13), and "Falls prevention initiatives" (C-14).

## Correlations between three perspectives in experts/safety managers

The correlations between the three perspectives of the experts are shown in Table 5A. There was a positive correlation between past contribution and current dissemination (r = 0.920, p < 0.001); however, there was no significant correlation between past contribution and future priority (r = -0.131, p = 0.474) or current dissemination and future priority (r = -0.273, p = 0.131).

The correlations for the safety managers are presented in Table 5B. Positive correlations were found between past contribution and future priority (r = 0.812, p < 0.001), current dissemination and future priority (r = 0.691, p < 0.001), and past contribution and current dissemination (r = 0.885, p < 0.001).

## Discussion

Conducting a secondary data analysis of the Delphi survey for Japanese experts and a nationwide questionnaire survey for safety managers in hospitals, we revealed the priority interventions for patient safety, together with the difference between the priority of safety managers

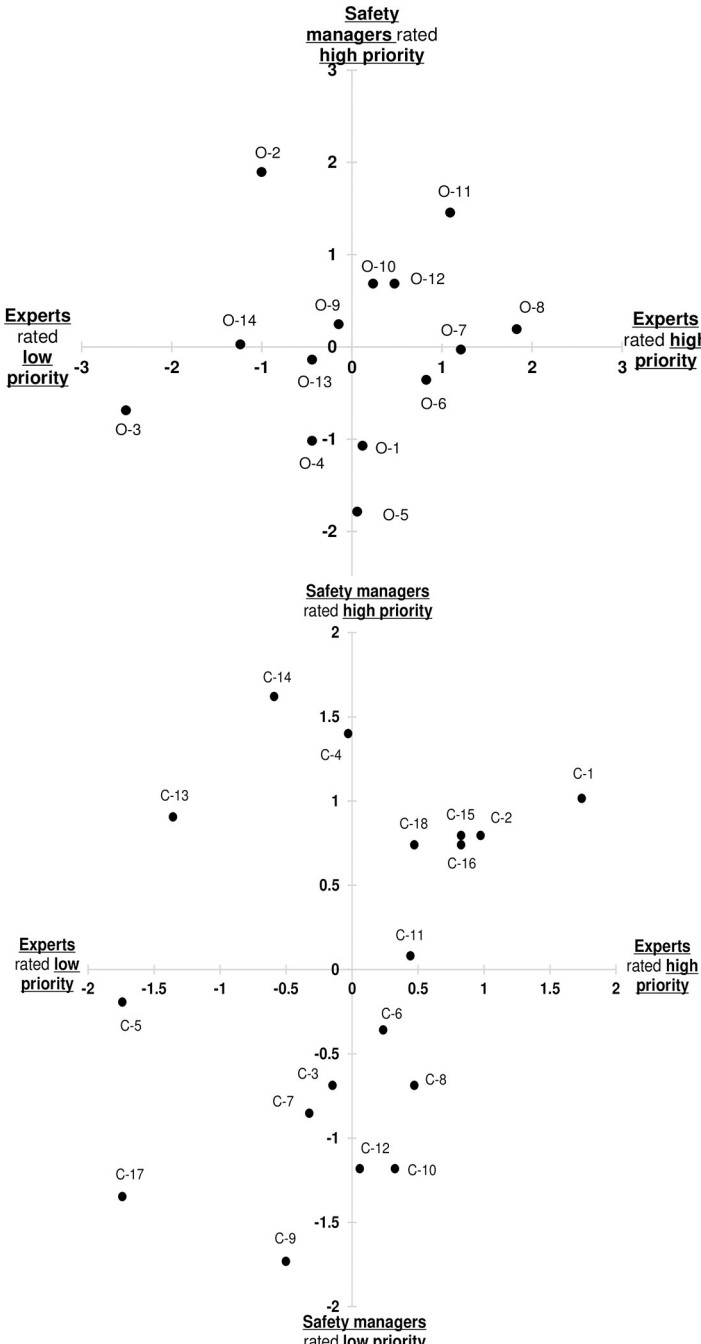

**Fig 1.** a. Priority of patient safety interventions (Organizational level). b. Priority of patient safety interventions (Clinical level).

and those of experts, and the relationship of future priority with past contribution and current dissemination.

In this study, as for the correlation of assessment for 32 patient safety interventions between experts and safety managers, positive correlations were observed in past contribution and current dissemination, but not in future priority. In experts, no significant correlation was observed between past contribution and future priority or between current dissemination and

**Table 4.**

**a. The mean values of 14 organizational-level interventions.**

| | | | n | Contribution* | Dissemination* | Priority* |
|---|---|---|---|---|---|---|
| O-1 | Clinical governance frameworks and systems for patient safety | Experts | 24 | -0.76 | -0.72 | 0.12 |
| | | Safety managers | 490 | -1.55 | -1.43 | -1.07 |
| O-2 | Clinical incident reporting and management system | Experts | 24 | 1.41 | 1.69 | -1.00 |
| | | Safety managers | 517 | 1.56 | 1.60 | 1.90 |
| O-3 | Integrated patient complaint- and incident-reporting | Experts | 23 | -1.02 | -0.91 | -2.51 |
| | | Safety managers | 515 | -0.56 | 1.33 | -0.69 |
| O-4 | Monitoring and feedback of patient safety indicators | Experts | 23 | -1.27 | -1.18 | -0.44 |
| | | Safety managers | 509 | -1.11 | -0.55 | -1.02 |
| O-5 | Patient-engagement initiatives | Experts | 23 | -1.79 | -1.82 | 0.06 |
| | | Safety managers | 501 | -2.03 | -1.26 | -1.79 |
| O-6 | Clinical communication protocols and training | Experts | 23 | -0.78 | -1.18 | 0.83 |
| | | Safety managers | 506 | -1.27 | -1.41 | -0.36 |
| O-7 | Digital technology solutions to improve safety | Experts | 24 | -0.52 | -0.04 | 1.21 |
| | | Safety managers | 508 | -0.60 | -0.30 | -0.03 |
| O-8 | Human resources interventions | Experts | 24 | -1.13 | -1.95 | 1.83 |
| | | Safety managers | 501 | -0.76 | -0.66 | 0.19 |
| O-9 | Building a positive safety culture | Experts | 24 | -0.64 | -0.39 | -0.15 |
| | | Safety managers | 514 | -0.60 | -0.80 | 0.25 |
| O-10 | Infection detection, reporting, and surveillance systems | Experts | 24 | 1.52 | 1.08 | 0.24 |
| | | Safety managers | 502 | 0.88 | 0.98 | 0.69 |
| O-11 | Hand hygiene initiatives | Experts | 24 | 0.80 | 0.75 | 1.09 |
| | | Safety managers | 512 | 1.52 | 1.49 | 1.46 |
| O-12 | Antimicrobial stewardship | Experts | 24 | -0.41 | -0.47 | 0.47 |
| | | Safety managers | 499 | 0.68 | 0.88 | 0.69 |
| O-13 | Blood and blood product management protocols | Experts | 23 | 1.47 | 1.33 | -0.44 |
| | | Safety managers | 494 | 0.84 | 1.02 | -0.14 |
| O-14 | Medical equipment sterilization protocols | Experts | 24 | 1.26 | 1.44 | -1.24 |
| | | Safety managers | 509 | 0.88 | 1.10 | 0.03 |

**b. The mean values of 18 clinical-level interventions.**

| | Interventions | | n | Contribution* | Dissemination* | Priority* |
|---|---|---|---|---|---|---|
| C-1 | Medication management / reconciliation protocols | Experts | 23 | -1.27 | -0.64 | 1.74 |
| | | Safety managers | 507 | 0.20 | -0.22 | 1.02 |
| C-2 | Transcribing error systems and protocols | Experts | 24 | 0.08 | 0.04 | 0.97 |
| | | Safety managers | 510 | 0.64 | 0.02 | 0.80 |
| C-3 | Smart infusion pumps and drug administration systems | Experts | 24 | -0.29 | -0.72 | -0.15 |
| | | Safety managers | 493 | 0.00 | -0.38 | -0.69 |
| C-4 | Aseptic technique protocols and barrier precautions | Experts | 24 | 1.84 | 1.08 | -0.03 |
| | | Safety managers | 502 | 1.79 | 1.43 | 1.40 |
| C-5 | Urinary catheter use and insertion protocols | Experts | 23 | 0.23 | 0.90 | -1.74 |
| | | Safety managers | 503 | 0.60 | 0.89 | -0.19 |
| C-6 | Central venous catheter insertion protocols | Experts | 24 | 1.15 | 0.48 | 0.24 |
| | | Safety managers | 477 | -0.12 | -0.34 | -0.36 |
| C-7 | Ventilator-associated pneumonia minimisation protocols | Experts | 23 | 0.08 | -0.02 | -0.32 |
| | | Safety managers | 467 | -0.28 | -0.54 | -0.85 |
| C-8 | Procedural / surgical checklists | Experts | 24 | 0.98 | 0.83 | 0.47 |
| | | Safety managers | 439 | -0.08 | -0.48 | -0.69 |

*(Continued)*

**Table 4.** (Continued)

| C-9 | Operating room integration and display technology | Experts | 24 | -0.64 | -0.91 | -0.50 |
|-----|---------------------------------------------------|---------|----|-------|-------|-------|
| | | Safety managers | 409 | -1.11 | -1.21 | -1.73 |
| C-10 | Peri-operative medication protocols | Experts | 23 | -0.03 | -0.20 | 0.32 |
| | | Safety managers | 417 | -0.72 | -1.16 | -1.18 |
| C-11 | Venous thromboembolism (VTE) prevention protocols | Experts | 23 | 0.72 | 0.90 | 0.44 |
| | | Safety managers | 478 | 0.12 | 0.07 | 0.08 |
| C-12 | Clinical care standards | Experts | 23 | 0.23 | -0.02 | 0.06 |
| | | Safety managers | 422 | -0.76 | -1.33 | -1.18 |
| C-13 | Pressure injury (ulcer) prevention protocols | Experts | 23 | 0.60 | 0.90 | -1.36 |
| | | Safety managers | 510 | 1.36 | 1.41 | 0.91 |
| C-14 | Falls prevention initiatives | Experts | 23 | -0.01 | 0.63 | -0.59 |
| | | Safety managers | 519 | 1.28 | 1.28 | 1.62 |
| C-15 | Acute delirium & cognitive impairment management initiatives | Experts | 23 | -1.07 | -1.18 | 0.83 |
| | | Safety managers | 492 | -0.48 | -0.66 | 0.80 |
| C-16 | Response to clinical deterioration | Experts | 23 | -0.81 | -0.83 | 0.83 |
| | | Safety managers | 487 | -0.32 | -0.64 | 0.74 |
| C-17 | Patient hydration and nutrition standards | Experts | 23 | -1.07 | -0.29 | -1.74 |
| | | Safety managers | 481 | -1.07 | -0.66 | -1.35 |
| C-18 | Patient identification and procedure matching protocols | Experts | 24 | 1.18 | 1.44 | 0.47 |
| | | Safety managers | 476 | 1.08 | 0.52 | 0.74 |

*: The scores were standardized into z-scores for adjusting the variables in the evaluation by experts and safety managers.

future priority. For safety managers, the evaluations of past contribution, current dissemination, and future priority were in the same direction. These results suggest that safety managers are likely to evaluate future priority based on past contribution and current dissemination; however, experts are likely to evaluate future priority based on other factors.

**Table 5.**

**a. Correlations between three perspective scores in experts.**

| | | Contribution | Dissemination | Priority |
|---|---|---|---|---|
| Contribution | r | 1.000 | | |
| | p | | | |
| Dissemination | r | 0.920 | 1.000 | |
| | p | $p < 0.001$ | | |
| Priority | r | -0.131 | -0.273 | 1.000 |
| | p | $p = 0.474$ | $p = 0.131$ | |

**b. Correlations between three perspective scores in safety managers.**

| | | Contribution | Dissemination | Priority |
|---|---|---|---|---|
| Contribution | r | 1.000 | | |
| | p | | | |
| Dissemination | r | 0.885 | 1.000 | |
| | p | $p < 0.001$ | | |
| Priority | r | 0.812 | 0.691 | 1.000 |
| | p | $p < 0.001$ | $p < 0.001$ | |

Abbreviation: r = Pearson's correlation coefficients.

**Table 6. A summary list of interventions according to the priorities given by experts and safety managers.**

| High priority by safety managers / Low priority by experts | | High priority by experts / Low priority by safety manages | |
|---|---|---|---|
| Organizational level | | Organizational level | |
| O-2 | Clinical incident reporting and management system | O-1 | Clinical governance frameworks and systems for patient safety |
| O-9 | Building a positive safety culture | O-5 | Patient-engagement initiatives |
| O-14 | Medical equipment sterilization protocols | O-6 | Clinical communication protocols and training |
| Clinical level | | Clinical level | |
| C-4 | Aseptic technique protocols and barrier precautions | C-6 | Central venous catheter insertion protocols |
| C-13 | Pressure injury (ulcer) prevention protocols | C-8 | Procedural / surgical checklists |
| C-14 | Falls prevention initiatives | C-10 | Peri-operative medication protocols |
| | | C-12 | Clinical care standards |

The interventions that were given high priority by safety managers, but low priority by experts were shown in the left column of Table 6. These interventions are relatively easy to conduct at the hospital or clinical level with ingenuity in the clinical setting because their evaluations of past contribution and current dissemination are high. "Building a positive safety culture" (O-9) has a lower evaluation of past contribution and current dissemination by patient safety managers. The importance of safety culture has been emphasized in other industries [27] and was also specified in the General Policy for Medical Safety published by the Ministry of Health, Labour, and Welfare, which defined the main framework of patient safety policies in Japan [28]. A previous study translated a hospital survey on patient safety culture (HSOPS) developed by the Agency for Healthcare Research and Quality in the United States into Japanese and evaluated its validity and applicability [29]. The Japan Council for Quality Health Care, which is a hospital accreditation body in Japan, started the benchmarking project using the Japanese version of the HSOPS to support hospitals in assessing their patient safety culture in 2020, and about 70 hospitals are participating in the project [30]. Experts might consider safety culture as a goal rather than an intervention, since the General Policy for Medical Safety stipulates that the ultimate goal of patient safety measures is to foster a patient safety culture. Interest in the survey on patient safety culture has been increasing and is becoming widely accepted, and the priority of "building a positive safety culture" might be highly evaluated by safety managers.

The interventions that were given high priority by experts, but low priority by safety managers were shown in the right column of Table 6. All these interventions had low scores for past contribution and current dissemination, as evaluated by safety managers. Our previous study on Japanese patient safety experts suggested that experts are likely to evaluate the priority of patient safety interventions with the expected impact in the future [20]. These results suggest that safety managers were likely to evaluate the priority of patient safety interventions in terms of which interventions were effective and important in the past, while experts were likely to evaluate what was lacking now for further improvement in patient safety. This difference seemed to be caused by that expert's assessment might include a healthcare system and policy perspective, while safety manager's assessment might be based on the needs and resources on their clinical and organizational settings. Measures in healthcare systems and policies, such as providing incentives or education for these interventions that were given high priority by

experts, but low priority by safety managers, should be investigated to facilitate their implementation.

In this study, we clarified high-priority interventions based on two surveys. However, it is unclear what kind of hospital can implement these interventions. Examining the relationship between hospital characteristics and reporting culture, a previous study revealed that large acute care hospitals with critical care centers had more voluntary in-hospital reports [23]. Clarifying the characteristics of hospitals that could implement interventions that were evaluated as high priority in this study would facilitate the dissemination of high-priority interventions.

## Limitations of this study

This study had some limitations. Only 32 patient safety interventions among many interventions were evaluated because comparability with other studies, including the OECD report, was emphasized. Safety managers in hospitals who were active in patient safety activities could more likely respond to the questionnaire survey, and the survey data might lack representativeness. Whether the safety managers' evaluation and priority setting of interventions may vary according to the activity of hospitals is unknown. The results of this study should be applied to other countries and regions with caution because this study was based on the results of surveys of patient safety experts and patient safety managers in Japan, and past contribution and current dissemination of each intervention may be different. However, this method of assessing intervention priorities and investigating influencing factors may also be applicable to future surveys in other countries and regions.

## Conclusion

We found that there were positive correlations in the score of past contribution and current dissemination of patient safety interventions evaluated by experts and safety managers, but future priorities were different. In experts, there was no significant correlation between past contribution and future priority or between current dissemination and future priority. For safety managers, the evaluations of past contribution, current dissemination, and future priority were in the same direction. The results of this study suggest that experts are likely to evaluate priority based on what is lacking now and would be needed in the future, although safety managers are likely to evaluate priority based on past contribution and current dissemination. In health policymaking, promotion of patient safety interventions that were given high priority by experts, but low priority by safety managers should be considered with possible incentives.

## Supporting information

**S1 Table. Questionnaire items of evaluation on organizational and clinical level interventions for patient safety in the Delphi survey.**
(PDF)

**S2 Table. Questionnaire items of evaluation on organizational and clinical level interventions for patient safety in the nationwide survey of patient safety managers in hospitals.**
(PDF)

## Author Contributions

**Conceptualization:** Ryosuke Hayashi, Yosuke Hatakeyama, Ryo Onishi, Kanako Seto, Kunichika Matsumoto, Tomonori Hasegawa.

**Data curation:** Ryosuke Hayashi, Yosuke Hatakeyama.

**Formal analysis:** Ryosuke Hayashi, Yosuke Hatakeyama.

**Funding acquisition:** Tomonori Hasegawa.

**Investigation:** Ryosuke Hayashi.

**Methodology:** Ryosuke Hayashi.

**Project administration:** Ryosuke Hayashi.

**Supervision:** Tomonori Hasegawa.

**Visualization:** Ryosuke Hayashi.

**Writing – original draft:** Ryosuke Hayashi.

**Writing – review & editing:** Ryosuke Hayashi, Yosuke Hatakeyama, Ryo Onishi, Kanako Seto, Kunichika Matsumoto, Tomonori Hasegawa.

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
