## [Decision Letter · Decision Letter 0]

8 Aug 2022

PONE-D-22-13129Difference in prioritization of patient safety interventions between experts and patient safety managers in JapanPLOS ONE

Dear Dr. Hasegawa,

Thank you for submitting your manuscript to PLOS ONE. After careful consideration, we feel that it has merit but does not fully meet PLOS ONE’s publication criteria as it currently stands. Therefore, we invite you to submit a revised version of the manuscript that addresses the points raised during the review process.

We look forward to receiving your revised manuscript.

Kind regards,

Jibril Mohammed, BSc, MSc, PhD

Academic Editor

PLOS ONE

Journal Requirements:

Additional Editor Comments (if provided):

This paper has some potential for publication. However, there is a need to improve on the writing style as well as statistical analyses. Please pay a careful attention to the comments of reviewer 1.

Reviewers' comments:

Reviewer's Responses to Questions

**Comments to the Author**

1. Is the manuscript technically sound, and do the data support the conclusions?

Reviewer #1: Yes

Reviewer #2: Yes

2. Has the statistical analysis been performed appropriately and rigorously? 

Reviewer #1: I Don't Know

Reviewer #2: Yes

3. Have the authors made all data underlying the findings in their manuscript fully available?

Reviewer #1: Yes

Reviewer #2: Yes

4. Is the manuscript presented in an intelligible fashion and written in standard English?

Reviewer #1: Yes

Reviewer #2: Yes

5. Review Comments to the Author

Reviewer #1: Thank you for the opportunity to review this manuscript which is a secondary analysis of two datasets looking at the patient safety interventions prioritised by experts and patient safety managers. The paper is mostly very well-written and presents an interesting opportunity to compare two influential groups in patient safety, considering the nuances and explanations for differences and similarities. Despite this potential, I do have some major concerns regarding the manuscript that need to be addressed by the authors. These are provided below.

1. Enhance the rationale for the study to further justify 1. Why these groups may differ in priorities, and 2. What the significance of this is.

2. Develop theoretically and empirically the concepts under investigation particularly the notion of the three perspectives: past contribution, present dissemination, and future priority. These need to be defined (including in the abstract) to be meaningful as well as providing more detail on the wording of questions about these perspectives. Ideally, and touching on previous comment related to rationale, these concepts would be considered in the introduction since there is an implicit assumption in this study that past contribution might affect future priority etc. I’d suggest exploring these concepts and their relationship to each other theoretically in addition to describing in the methods how they were defined/assessed, e.g., consider past literature to support these ideas, make hypotheses.

I have a few other concerns about the comparison of these two datasets

3. In methods, explain what makes someone an expert, especially given this group also contains safety managers.

4. Also clarify whether the study was positioned differently for these different groups or whether the wording of questions was slightly different e.g., “for your hospital” for patient safety managers, vs. “national priorities” for patient safety experts. I’d suggest including both survey forms as appendices.

5. Use of a Delphi alongside a traditional survey for comparison throws up a number of issues that need to be addressed: 1. What round of the Delphi responses are the authors using in the analysis? 2. If they are using round three, the authors should consider whether they are comparing experts vs. managers – arguably they are comparing a group that has refined their responses in light of controlled feedback toward a consensus, as compared with another group who has responded to a one off survey.

Although well-written, there are few instances where wording is overly complicated. For example:

1. Saying “high and low priority by experts and safety managers, respectively” is more complicated than saying “high priority by experts, but low priority by safety managers” and this is a recurrent issue. In addition to revising this wording throughout, it might be useful to create a box or table that summarises the differences e.g., low priority by experts/high priority by managers in one column and high priority by experts/low priority by managers in another. Doing so would also allow some trimming in the Discussion, where these Results are currently repeated. I’d suggest also being explicit on cut offs for when something is deemed low vs high priority.

Reviewer #2: This paper was well organized and clearly articulated the similarities and differences between experts and patient safety managers in regard to prioritization.

Here are a few points for consideration:

1. Dissemination and prioritization can be interpreted differently by various respondents. Were those that participated in the Delphi panel provided definitions of dissemination and prioritization?

2. In the analysis the authors state that "Except for the 10 system-level interventions not asked about dissemination in the questionnaire survey of safety managers..." Is there a rationale for why these 10 system level interventions were not included?

3. In the discussion, it would be helpful to more clearly articulate what the potential reasons and implications are for differences in prioritization among experts and patient safety managers.

6. PLOS authors have the option to publish the peer review history of their article (what does this mean?). If published, this will include your full peer review and any attached files.

Reviewer #1: No

Reviewer #2: No

---

## [Author Response · Author response to Decision Letter 0]

12 Sep 2022

11/09/2022

Dear Jibril Mohammed, BSc, MSc, PhD

We thank you for the careful evaluation of our manuscript and allowing us to revise. We appreciate your valuable comments and we would like to respond as following:

1

The PLOS ONE style templates can be found at 

and 

Answer:

We checked the format samples and made some corrections to meet PLOS ONE's style requirements, including those for file naming.

2

 Please provide additional details regarding participant consent.

In the ethics statement in the Methods and online submission information, please ensure that you have specified what type you obtained (for instance, written or verbal, and if verbal, how it was documented and witnessed).

If your study included minors, state whether you obtained consent from parents or guardians.

If the need for consent was waived by the ethics committee, please include this information.

Answer:

Both surveys did not include minors. We added following explanations.

> P. 6, L. 97-98

All participants of both surveys were informed about the research objective and the policy of data confidentiality and anonymity.

3

 In your Data Availability statement, you have not specified where the minimal data set underlying the results described in your manuscript can be found.

PLOS defines a study's minimal data set as the underlying data used to reach the conclusions drawn in the manuscript and any additional data required to replicate the reported study findings in their entirety. 

All PLOS journals require that the minimal data set be made fully available. 

For more information about our data policy, please see 

http://journals.plos.org/plosone/s/data-availability.

Answer:

We performed a secondary data analysis of two past surveys. Data were provided for secondary analysis but were not permitted to be published. The aggregated results of the data used for analysis are shown in Table 4 in the paper.

Reviewer 1:

Comment 1

Enhance the rationale for the study to further justify 1. Why these groups may differ in priorities, and 2. What the significance of this is.

Answer:

We corrected and added following explanations.

> P. 5, L. 77-87

Practitioners and patient safety managers are likely to provide patient safety interventions based on the needs and resources on their clinical and organizational settings. Patient safety experts might assess the priority interventions from the perspective of a healthcare system and policy. Therefore, the intervention priorities of safety managers might be different from those of experts. Previous studies have reported the priorities and determinants of patient safety interventions at the clinical level [21,22]. However, few studies have shown differences in intervention priorities between experts and safety managers. By clarifying the differences in prioritization and influential factors, we could identify interventions to be promoted through the support of healthcare system and policies.

Also, we corrected the words related to this point.

> P. 7, L. 118

> P. 8, L. 139

> P. 13, L. 204

> P. 22, L. 249

current present

Comment 2

Develop theoretically and empirically the concepts under investigation particularly the notion of the three perspectives: past contribution, present dissemination, and future priority.

These need to be defined (including in the abstract) to be meaningful as well as providing more detail on the wording of questions about these perspectives.

Ideally, and touching on previous comment related to rationale, these concepts would be considered in the introduction since there is an implicit assumption in this study that past contribution might affect future priority etc.

I’d suggest exploring these concepts and their relationship to each other theoretically in addition to describing in the methods how they were defined/assessed, e.g., consider past literature to support these ideas, make hypotheses.

Answer:

We prepared both survey forms focusing on items used in this study as supporting information to provide more detail on the wording of questions about these perspectives.

S1 Table. Questionnaire items of evaluation on organizational and clinical level interventions for patient safety in the Delphi survey.

S2 Table. Questionnaire items of evaluation on organizational and clinical level interventions for patient safety in the nationwide survey of patient safety managers in hospitals.

And we added following explanations.

> P. 6, L. 96-97

Parts of both questionnaire items used in this study were shown in S1 and S2 Tables.

Additionally, we corrected and added following explanations.

> P. 4-5, L. 63-76

In this study, the questionnaire consisted of 42 interventions based on the OECD report, and 6 perspectives for assessing the importance of interventions in the past (contribution), current (dissemination), and future (impact, cost, urgency, and priority). They reported that the priority of patient safety interventions had a positive relationship with the future impact and a negative relationship with current dissemination. These results suggested that experts gave high priority to interventions that were expected to be effective in the future and low priority to interventions that were already disseminated. It seemed to be important for policymakers and hospital administrators to consider the status of the medical system, the medical policies that had been taken thus far, and the circumstances that were important in setting the priority of patient safety interventions. These OECD report and previous study suggested that future priorities for interventions might be influenced not only by expected future impact, but also by past contribution and current dissemination.

Comment 3

 In methods, explain what makes someone an expert, especially given this group also contains safety managers.

Answer:

To clarify the meaning of this matter, we added following explanations.

> P. 6, L. 108-109

In the Delphi survey, the criteria for experts were to be actively involved in academic activities such as academic conferences or writing papers of patient safety. 

Comment 4

Also clarify whether the study was positioned differently for these different groups or whether the wording of questions was slightly different e.g., “for your hospital” for patient safety managers, vs. “national priorities” for patient safety experts. I’d suggest including both survey forms as appendices.

Answer:

We prepared both survey forms focusing on items used in this study as supporting information to provide more detail on the wording of questions about these perspectives.

S1 Table. Questionnaire items of evaluation on organizational and clinical level interventions for patient safety in the Delphi survey.

S2 Table. Questionnaire items of evaluation on organizational and clinical level interventions for patient safety in the nationwide survey of patient safety managers in hospitals.

And we added following explanations.

> P. 6, L. 96-97

Parts of both questionnaire items used in this study were shown in S1 and S2 Tables.

Additionally, we added the explanation of wording of the questionnaires.

> P. 7-8, L. 133-138

Using the same wording as the Delphi survey for experts, respondents were asked to rate 42 interventions on a 5-point scale from two perspectives: past contribution to patient safety (contribution: 1: small to 5: large) and priority for future implementation (priority: 1: low to 5: high). They were also asked whether 14 interventions at the organizational level and 18 at the clinical level (totaling 32 interventions) were implemented in their hospitals with the wording of “at your hospital”.

Comment 5

 Use of a Delphi alongside a traditional survey for comparison throws up a number of issues that need to be addressed: 1. What round of the Delphi responses are the authors using in the analysis? 2. If they are using round three, the authors should consider whether they are comparing experts vs. managers – arguably they are comparing a group that has refined their responses in light of controlled feedback toward a consensus, as compared with another group who has responded to a one off survey.

Answer:

We added following explanations.

> P. 8-9, L. 155-160

As data of experts for current dissemination and future priority, we used the results of round three converged through the Delphi process for analyzing the representative perspectives of patient safety experts in Japan. We included the results of round one on past contribution, as they were only asked at round one. The perspectives of safety managers were varied depending on their own circumstances, there was no need to converge them for the analysis.

Comment 6

Although well-written, there are few instances where wording is overly complicated. For example:

1. Saying “high and low priority by experts and safety managers, respectively” is more complicated than saying “high priority by experts, but low priority by safety managers” and this is a recurrent issue. In addition to revising this wording throughout, it might be useful to create a box or table that summarises the differences e.g., low priority by experts/high priority by managers in one column and high priority by experts/low priority by managers in another. Doing so would also allow some trimming in the Discussion, where these Results are currently repeated. I’d suggest also being explicit on cut offs for when something is deemed low vs high priority.

Answer:

To simplify the expression, the expressions regarding the priority in "Results" are modified as follows.

> P. 3, L. 40-41

> P. 14, L. 208

> P. 14, L. 211-212

high priority by experts, but low priority by safety managers

> P. 14, L. 216

> P. 14, L. 220

high priority by safety managers, but low priority by experts

We added two tables and corrected following explanations.

Table 6. The interventions that were given high priority by safety managers, but low priority by experts

Table 7. The interventions that were given high priority by experts, but low priority by safety managers

> P. 23, L. 259-260

The interventions that were given high priority by safety managers, but low priority by experts were shown in Table 6.

The interventions that were given low and high priority by experts and safety managers, respectively, were “Clinical incident reporting and management system” (O-2), “Medical equipment sterilization protocols” (O-14) in the organization level, “Aseptic technique protocols and barrier precautions” (C-4), “Pressure injury (ulcer) prevention protocols” (C-13), and “Falls prevention initiatives” (C-14) in clinical level.

> P. 25, L. 287-288

The interventions that were given high priority by experts, but low priority by safety managers were shown in Table 7.

The interventions that were given high and low priority by experts and safety managers, respectively, were “Clinical governance frameworks and systems for patient safety” (O-1), “Patient-engagement initiatives” (O-5), and “Clinical communication protocols and training” (O-6) in the organizational level, “Central venous catheter insertion protocols” (C-6), “Procedural / surgical checklists” (C-8), “Peri-operative medication protocols” (C-10), and “Clinical care standards” (C-12) in the clinical level.

To clarify the definition of high and low priority, we added following explanations.

> P. 14, L. 207

We defined scores > 0 as 'high' and < 0 as 'low'.

Reviewer 2:

Comment 1

Dissemination and prioritization can be interpreted differently by various respondents. Were those that participated in the Delphi panel provided definitions of dissemination and prioritization?

Answer:

We prepared both survey forms focusing on items used in this study as supporting information to provide more detail on the wording of questions about these perspectives.

S1 Table. Questionnaire items of evaluation on organizational and clinical level interventions for patient safety in the Delphi survey.

S2 Table. Questionnaire items of evaluation on organizational and clinical level interventions for patient safety in the nationwide survey of patient safety managers in hospitals.

And we added following explanations.

> P. 6, L. 96-97

Parts of both questionnaire items used in this study were shown in S1 and S2 Tables.

Comment 2

 In the analysis the authors state that "Except for the 10 system-level interventions not asked about dissemination in the questionnaire survey of safety managers..." Is there a rationale for why these 10 system level interventions were not included?

Answer:

To clarify the meaning of this matter, we corrected following explanations.

> P. 8, L. 143-148

We assessed the mean values of 32 interventions (Table1), consisting of 14 organizational-level interventions and 18 clinical-level interventions in past contribution, current dissemination, and priority for future implementation. The 10 system-level interventions were not included in the questionnaire for safety managers because these interventions involved the entire national healthcare system and could not be implemented in each hospital.

Except for the 10 system-level interventions not asked about dissemination in the questionnaire survey of safety managers, we assessed the mean values of 32 interventions (Table1), consisting of 14 organizational-level interventions and 18 clinical-level interventions in past contribution, present dissemination, and priority for future implementation.

Comment 3

 In the discussion, it would be helpful to more clearly articulate what the potential reasons and implications are for differences in prioritization among experts and patient safety managers.

Answer:

We added following explanations.

> P. 226, L. 300-303

This difference seemed to be caused by that expert's assessment might include a healthcare system and policy perspective, while safety manager's assessment might be based on the needs and resources on their clinical and organizational settings.

---

## [Editor Report · Decision Letter 1]

29 Nov 2022

PONE-D-22-13129R1Difference in prioritization of patient safety interventions between experts and patient safety managers in JapanPLOS ONE

Dear Dr. Hasegawa,

Thank you for submitting your manuscript to PLOS ONE. After careful consideration, we feel that it has merit but does not fully meet PLOS ONE’s publication criteria as it currently stands. Therefore, we invite you to submit a revised version of the manuscript that addresses the points raised during the review process.

We look forward to receiving your revised manuscript.

Kind regards,

Keiko Nakamura

Academic Editor

PLOS ONE

Journal Requirements:

Additional Editor Comments:

The manuscript was reasonably revised according to the reviewers' comments with additional tables and supplementary tables.

Please pay attention to the following points and revise the manuscript accordingly.

Table 6 and Table 7 should not duplicate the results presented in Table 4a and Table 4b.

Combine Tables 6 and 7 and list the code (for example “O-2”) and interventions (for example “Clinical incident reporting and management system) showing “low priority by experts/high priority by managers” in one column (for example left column) and those showing “high priority by experts/low priority by manages” in another column (for example right column). The number of respondents and scores were presented in Table 4a and Table 4b, therefore these are not needed. A potential title of the updated Table 6 is “A summary list of interventions according to the priorities given by experts and safety managers”.

Confusing sentences are still remain.

“experts and safety managers had given high and low priority, respectively” (lines 284-285)

“experts and safety managers had given high and low priority” (line 322)

Clearer expressions are needed.
---

## [Author Response · Author response to Decision Letter 1]

27 Dec 2022

27/12/2022

Dear Dr. Keiko Nakamura,

We thank you for the careful evaluation of our manuscript and allowing us to revise. We appreciate your valuable comments and we would like to respond as following:

Answer:

We rechecked all our reference lists and determined everything was fine.

Additional Editor Comments 1

Table 6 and Table 7 should not duplicate the results presented in Table 4a and Table 4b.

Combine Tables 6 and 7 and list the code (for example “O-2”) and interventions (for example “Clinical incident reporting and management system) showing “low priority by experts/high priority by managers” in one column (for example left column) and those showing “high priority by experts/low priority by manages” in another column (for example right column). The number of respondents and scores were presented in Table 4a and Table 4b, therefore these are not needed. A potential title of the updated Table 6 is “A summary list of interventions according to the priorities given by experts and safety managers”.

Answer:

Following your suggestion, we combined Table 6 and Table 7 into new Table 6, revised the structure of Table 6, and changed the title of it.

Table 6. A summary list of interventions according to the priorities given by experts and safety managers

High priority by safety managers / Low priority by experts High priority by experts / Low priority by safety manages

Organizational level Organizational level

O-2 Clinical incident reporting and management system O-1 Clinical governance frameworks and systems for patient safety

O-9 Building a positive safety culture O-5 Patient-engagement initiatives

O-14 Medical equipment sterilization protocols O-6 Clinical communication protocols and training

Clinical level Clinical level

C-4 Aseptic technique protocols and barrier precautions C-6 Central venous catheter insertion protocols

C-13 Pressure injury (ulcer) prevention protocols C-8 Procedural / surgical checklists

C-14 Falls prevention initiatives C-10 Peri-operative medication protocols

　 　 C-12 Clinical care standards

And, we corrected following explanations in the body.

> P. 22, L. 251

The interventions that were given high priority by safety managers, but low priority by experts were shown in left column of Table 6.

> P. 26, L. 277

The interventions that were given high priority by experts, but low priority by safety managers were shown in right column of Table 67.

Additional Editor Comments 2

Confusing sentences are still remain.

“experts and safety managers had given high and low priority, respectively” (lines 284-285)

“experts and safety managers had given high and low priority” (line 322)

Clearer expressions are needed.

Answer:

To simplify, we corrected following explanations.

> P. 26, L. 288-289

were given high priority by experts, but low priority by safety managers

experts and safety managers had given high and low priority, respectively

> P. 29, L. 328-329

were given high priority by experts, but low priority by safety managers

experts and safety managers had given high and low priority

We have rechecked our entire manuscript and aligned columns in Table 2b.

Table 2a. Baseline characteristics (Experts). 

 n %

Experts 24 

Domain 

Representative of nationwide organization related to patient safety 2 8.3

Hospital administrator 5 20.8

Patient safety manager 7 29.2

Researcher of patient safety 8 33.3

Other 2 8.3

Profession 

Doctor 15 62.5

Nurse 4 16.7

Pharmacist 2 8.3

Others 3 12.5

Table 2b. Baseline characteristics (Safety managers). 

　 n %

Safety managers 603 

Acute care hospital 

< 100 beds 68 11.3

100–299 beds 178 29.5

≥ 300 beds 225 37.3

Chronic care hospital 

< 100 beds 29 4.8

≥ 100 beds 48 8.0

Psychiatric hospital 46 7.6

Other hospitals 9 1.5

At the time of submission, the editorial team pointed out the following corrections.

Please provide additional details regarding participant consent. 

In the Methods section, please ensure that you have specified (1) whether consent was informed and (2) what type you obtained (for instance, written or verbal).

If your study included minors, state whether you obtained consent from parents or guardians.

If the need for consent was waived by the ethics committee, please include this information.

Answer:

We did not obtain informed consent explicitly for participation in both investigations. However, all participants in both surveys were informed of the purposes of the surveys, data confidentiality and anonymity policies in the request letter. Additionally, taking part in the surveys was voluntary, not mandatory. 

Both surveys did not include minors. 

Based on our study protocols, the ethics committee did not request us to obtain participant consent.

We added following explanations.

> P. 6, L. 97-99

All participants of both surveys were informed about the research objective and the policy of data confidentiality and anonymity. Taking part in both surveys was voluntary, not mandatory. Therefore, we considered responses to surveys as consent to participate in the survey.

---

## [Editor Report · Decision Letter 2]

3 Jan 2023

Difference in prioritization of patient safety interventions between experts and patient safety managers in Japan

PONE-D-22-13129R2

Dear Dr. Hasegawa,

We’re pleased to inform you that your manuscript has been judged scientifically suitable for publication and will be formally accepted for publication once it meets all outstanding technical requirements.

Kind regards,

Keiko Nakamura

Academic Editor

PLOS ONE
---

## [Editor Report · Acceptance letter]

20 Feb 2023

PONE-D-22-13129R2 

Difference in prioritization of patient safety interventions between experts and patient safety managers in Japan 

Dear Dr. Hasegawa:

I'm pleased to inform you that your manuscript has been deemed suitable for publication in PLOS ONE. Congratulations! Your manuscript is now with our production department. 

Kind regards, 

on behalf of

Professor Keiko Nakamura 

Academic Editor

PLOS ONE